# Signal Quality Analysis for Long-Term ECG Monitoring Using a Health Patch in Cardiac Patients

**DOI:** 10.3390/s23042130

**Published:** 2023-02-14

**Authors:** Israel Campero Jurado, Ilde Lorato, John Morales, Lonneke Fruytier, Shavini Stuart, Pradeep Panditha, Daan M. Janssen, Nicolò Rossetti, Natallia Uzunbajakava, Irina Bianca Serban, Lars Rikken, Margreet de Kok, Joaquin Vanschoren, Aarnout Brombacher

**Affiliations:** 1Department of Mathematics and Computer Science, Eindhoven University of Technology, 5612 AZ Eindhoven, The Netherlands; 2Stichting IMEC Nederland, 5656 AE Eindhoven, The Netherlands; 3Department of Cardiology, Máxima Medical Center, De Run 4600, 5504 DB Veldhoven, The Netherlands; 4Holst Centre, TNO, Biomedical R&D, 5656 AE Eindhoven, The Netherlands; 5Department of Industrial Design, Eindhoven University of Technology, 5612 AZ Eindhoven, The Netherlands

**Keywords:** signal quality, electrocardiography, continuous monitoring, lifestyle, health patch, dry electrodes

## Abstract

Cardiovascular diseases (CVD) represent a serious health problem worldwide, of which atrial fibrillation (AF) is one of the most common conditions. Early and timely diagnosis of CVD is essential for successful treatment. When implemented in the healthcare system this can ease the existing socio-economic burden on health institutions and government. Therefore, developing technologies and tools to diagnose CVD in a timely way and detect AF is an important research topic. ECG monitoring patches allowing ambulatory patient monitoring over several days represent a novel technology, while we witness a significant proliferation of ECG monitoring patches on the market and in the research labs, their performance over a long period of time is not fully characterized. This paper analyzes the signal quality of ECG signals obtained using a single-lead ECG patch featuring self-adhesive dry electrode technology collected from six cardiac patients for 5 days. In particular, we provide insights into signal quality degradation over time, while changes in the average ECG quality per day were present, these changes were not statistically significant. It was observed that the quality was higher during the nights, confirming the link with motion artifacts. These results can improve CVD diagnosis and AF detection in real-world scenarios.

## 1. Introduction

Cardiovascular diseases (CVD) represent a major societal and economic burden [1], with specific cardiac events and symptoms such as embolic stroke, heart failure, or atrial fibrillation (AF) incorporating a large portion of healthcare costs [2,3], e.g., Eur 210 billion a year in Europe (2017) [4]. Considering AF alone, it is predicted that, by 2030, 15.9 million people in the US and 14–17 million people in Europe will suffer from AF [5]. Electrocardiography (ECG) is an important tool for the diagnosis and evaluation of patients with CVD. A 12-lead wired ECG system is currently used in clinical practice to provide the highest resolution and best visualization of the entire heart for clinical assessment [6]. An exercise ECG can provide additional information about the heart during stress. Therefore, long-term monitoring during physical activities in home settings will potentially lead to a more accurate diagnosis and prevention of CVD in specific populations [7].

The traditional 12-lead design is not well suited for prolonged patient monitoring at home, while dedicated 12-lead systems for short-term home monitoring start to populate the market [8,9,10], they are not designed to provide comfortable and long-term continuous monitoring in home settings. This problem has been addressed with the use of Holter monitors, which are wearable devices for heart rate monitoring that provide ECG signal acquisition for 24–72 h. Newer Holter devices can record for longer periods of up to 2 weeks [11]. However, the wired framework of the Holter and the use of wet gel electrodes increases the likelihood of measurement noise during motion [12,13] and reduces wear possibilities, e.g., it cannot be used during showering and necessitates replacement of the wet gel electrodes upon drying out of the gel.

In the past few years, sensors and technologies have been proposed to allow monitoring of patients at home and in everyday activities [14,15,16,17,18], with the majority being single-lead ECG solutions (also known as health patches), which in most cases rely on using wet resistive gel electrodes. Wet resistive gel electrodes will dehydrate after a few days of usage, eventually limiting long-term usability and reducing the quality of the signal over time [19,20]. Therefore, self-adhesive dry electrodes have been explored as alternatives since they can extend the duration of monitoring [21,22,23,24,25]. Studies show that dry electrodes behave better in terms of impedance and noise reduction performance than Ag/AgCl electrodes if properly shielded [26,27]. Within this study, a *Vital Signs patch* research platform (TNO Holst Centre. https://executivereport.holstcentre.com/innovation-updates/health-vitality/health-patch/ accessed on 20 January 2023) was used to ensure optimized skin–electrode contact for longer-term wear within a hybrid printed electronic format. Self-adhesive dry electrode technology was used and the overall patch was designed to be thin, durable, and lightweight for use in all daily activities.

Examples of using health patches for CVD monitoring include the detection of AF/flutter, for example in [28] a patch device was used for AF screening. AF was identified in 5.7% of patients who did not show any evidence of AF in normal ECG and Holter monitoring. Other studies with health patches were focused on comparing the performance of such devices against Holter monitoring systems in detecting CVD, where the single-lead health patches identified most of the arrhythmia events and showed a beat-per-minute correlation compared to the Holter system higher than 90% [14,15,16,17,18]. Health patches have also been reported to correctly identify more cardiac events than the Holter monitor in remote settings over a prolonged use period [29].

Signal quality monitoring is a paramount initial step for ECG analysis and cardiac markers or event detection. Currently, different metrics can be used for ECG signal quality assessment: signal-to-artifact ratio [13], signal-to-noise ratio [30], beats per minute (BPM) [31], and other derived features such as the slowest, the average, and the fastest heart rate (HR) [32]. However, previous studies did not assess the change in signal quality over time, which is an important factor in characterizing single-lead patches and possibly improving the materials used and the design.

The goal of this work is to analyze the quality of ECG signals recorded in a home setting during five consecutive days using a *Vital Signs patch* featuring self-adhesive dry electrodes technology. The changes in quality were studied between the days and between daytime and night. To address the goal we have selected a two-step approach. The first step is the comparison of four algorithms for signal quality assessment. As this requires a gold standard, a publicly available database with annotated quality was used. Afterward, the best-performing quality metric was applied to an ECG dataset collected on six patients with chronic coronary syndrome (CCS) who wore the *Vital Signs patch* for 5 days continuously.

The paper is organized as follows: Section 2 presents the datasets used and the experimental setup. Methods and algorithms used are detailed in Section 3, and Section 4 shows the results of our approach. Finally, Section 5 and Section 6 discuss the results and provide conclusions and future work suggestions.

## 2. Materials

Two datasets have been used in this work. The first is an open dataset with signals acquired from healthy adults and it was used to validate the proposed methodology. The second one includes data recorded using a *Vital Signs patch* based on self-adhesive dry electrodes from patients suffering from CCS.

### 2.1. Benchmarking Dataset

The *Brno University of Technology open ECG Quality Database* (QDB) [33] was selected as a dataset for benchmarking.

The dataset contains 18 ECG recordings, acquired with a minimum duration of 24 h from healthy adults, 6 men and 9 women between 21 and 83 years. The sampling frequency was 1000 Hz, and the data was recorded during daily activities. The ECG segments’ length can vary between a few seconds and minutes. These segments were annotated by three experts independently in three quality levels using the following criteria:Class 0: Onset–offset beat, P-peak, T-peak, and QRS complex are easily identified.Class 1: QRS complex duration is not distinguishable; PR interval and other points in the ECG are unclear.Class 2: The QRS complexes and peaks cannot be identified accurately.

The demographic information of the participants is summarized in Table 1.

### 2.2. Clinical Study

Raw ECG signals, raw accelerometer, and respiration were collected for five days from patients with CCS using a patch featuring self-adhesive dry electrode technology. The study was conducted following Medical Ethical Committee guidelines for non-WMO (not required by Medical Research Law) research. More details on the setup, the protocol, and the patients are described in this section.

#### 2.2.1. Device

The *Vital Signs patch* research platform, Figure 1, consists of a disposable sensing patch and a re-usable read-out module (see Figure 2, which shows assembled patch and marks a disposable part and a re-usable part). For this study, the platform was optimized with the disposable part containing three self-adhesive dry electrodes for acquiring single-lead ECG signals within a bipolar format with increased activity and longer duration in mind. These electrodes are integrated with screen-printed flexible and stretchable conducting circuitry, printed on conformable thermoplastic polyurethane (TPU) substrate (at the Holst Centre manufacturing facilities). Overall, a unique stretchable patch stack allows the skin to breathe and to adapt to the shape of the human body, with all material in contact with the body ISO 10993 certified for biocompatibility. The re-usable read-out unit interconnected with the flexible patch has accelerometer incorporated for monitoring of motion and activity. The read-out unit has a battery time of seven days, and a sampling rate of 256 Hz for the ECG signal. The device is also shower-proof to accommodate normal daily usage. The health patch research platform was developed within the TNO Holst Centre to evaluate wearable design considerations, including materials, printed electronics design, manufacturing, system architecture, and modularity of sensor platforms, for clinical diagnosis and monitoring possibilities.

#### 2.2.2. Dataset

Six adult cardiac patients with CCS were recruited using the patient database of the Cardiology department of the Máxima Medical Centre (MMC) in Veldhoven, The Netherlands. These participants underwent a routine exercise stress test (EST), performed on a bicycle. During this maximum test, the patient is encouraged to reach their maximum effort. The traditional 12-lead suction electrodes and the *Vital Signs patch* were positioned on the chest of the participants, with the *Vital Signs patch* placed in the 5th intercostal space, just below precordial leads V4–V6, as shown in Figure 3.

Data was captured for 10–15 min during the EST. Afterward, the patients were instructed to wear the patch continuously for five days. Additionally, each patient annotated their daily activities in a diary, including information such as the time and duration of walking or exercising. The detailed annotated activities can be seen in the Appendix A. The demographics of the patients are summarized in Table 2.

In total, 704 h should have been recorded cumulatively for all the patients. However, due to data loss probably caused by poor skin–electrode contact, 627 h are available for analysis. In Table 3, a summary of the amount of data per patient is provided, split between daytime (assumed from 7 a.m. to 11 p.m.) and night (assumed from 11 p.m. to 7 a.m.).

## 3. Methods

The processing was performed using Python programming language (Python Software Foundation 3.8.0) [34]. The analysis can be split into two main stages, as shown in Figure 4. In more detail:The first stage aimed to find an SQI metric. For this purpose, the benchmarking of four different SQIs on the QDB, introduced in Section 3.2, was performed.The second stage analyzed the quality of the data collected on cardiac patients. The best-performing SQI was applied on this dataset and the changes in quality during the days were analyzed.

### 3.1. Preprocessing

The preprocessing was common to both datasets. Firstly, the baseline drift noise was removed from the ECG signals with a 0.5 Hz high-pass Butterworth filter (order = 5). In the QDB dataset, where a single quality annotation is available for each ECG segment, such a segment was used for the quality analysis. This means that segments with different lengths were processed. The ECG signals of the cardiac dataset, instead, were further processed with a sliding window approach with a window size of 60 s and no overlap. Next, the R-peaks of the ECG signals were detected using the Python toolbox Neurokit [35], which uses an algorithm based on the slope of the absolute gradient of the ECG signal. There is a difference in population between the benchmarking dataset and the clinical dataset, i.e., healthy subjects and cardiac patients. Since abnormal beats can be expected in the cardiac population, preprocessing steps were added to handle the presence of irregular QRS complexes. In particular, the method described in [36] was applied. Each QRS complex was extracted using a window of 60 ms before and 60 ms after each R-peak. Figure 5a shows an example of this procedure. The variance of each segmented complex is computed, and thresholds are defined based on the 25th ($Q_{1}$) and 75th ($Q_{3}$) percentiles of the distribution of the variances, as well as on the interquartile range (IQR = Q3 − Q1). If a segmented QRS complex has a variance outside the interval between Q1−2.5*IQR and Q3+2.5*IQR, it is considered as irregular and is discarded from further analyses. An example of the distribution of the variances and the thresholds is shown in Figure 5b. The QRS complexes corresponding to a variance outside the defined range are discarded. Figure 5c,d show an example of the segmented QRS complexes within and outside the limits, respectively.

### 3.2. Signal Quality Indicators

The four SQIs compared in this work were partially selected based on promising results from the literature. They are easy to implement, reproducible, common approaches with available implementations and partially developed based on physiology. There are many notions of ECG signal quality, such as independent use of skewness, signal-to-noise ratio, and higher-order statistics, but they have been shown to be biased towards noise-free ECG signals and therefore cannot be used as a robust noise detection method [37].

#### 3.2.1. Sqi Based on the Average Qrs Complex ($Sqi_{Avg}$)

This SQI is part of Neurokit [35] and can be found in the source documentation of the library. This method, as other SQIs based on an average QRS complex used as a template, assumes most of the ECG signal’s segment under analysis to have a regular morphology. Therefore, the average complex will reflect the regular morphology and deviations from it can be identified. In particular, to calculate the $SQI_{avg}$ of a given ECG segment, the QRS complexes are isolated starting from the R-peak location. Afterwards, each QRS complex is standardized by subtracting the average QRS complex and then dividing by the standard deviation. The average of each standardized QRS complex is computed, and rescaling is applied so that the output is between 0 and 1, with 1 corresponding to the highest quality. This represents the SQI of each QRS complex; the average SQI for all the QRS complexes in the window was calculated and used as the main output. For more details, refer to the open source implementation [35].

#### 3.2.2. Sqi Based on Fuzzy Classifier ($Sqi_{Fuzzy}$)

The second method used in the benchmark is an implementation available in the Neurokit toolbox [35] based on the method originally proposed in [38]. This SQI uses a combination of different quality indexes. The data fusion approach is based on a fuzzy comprehensive evaluation. The Neurokit implementation [35] includes 3 popular statistical signal quality indexes: power spectrum distribution of QRS wave, baseline relative power, and kurtosis of the ECG signal. The overall output is then used to classify the signal into one of three quality levels: excellent, barely acceptable, or unacceptable.

#### 3.2.3. Sqi Based on Heart Rate Variability ($Sqi_{Hrv}$)

This SQI evaluates the statistical characteristics of the variations in heart rate. To compute it, the R-peaks in the ECG are first detected using the Neurokit toolbox [35]. It is known that populations with cardiac problems usually suffer from arrhythmias. These might bias the quality evaluation with $SQI_{hrv}$. The method explained in Section 3.1 removes some of the arrhythmias. However, an additional step was needed to mitigate the effect that irregular heartbeats might have in this SQI. For this, the R-peaks are adjusted as explained in [39]. This is a method based on time-varying thresholds that corrects for irregular heartbeats. Afterwards, the time differences between consecutive R-peaks (RRs) are calculated to produce a time series known as a tachogram. From the tachogram, one measurement that allows its characteristics to be evaluated is the square root of the mean squared differences of successive RR intervals (RMSSD) [40]. The RMSSD is defined as:(1)$$RMSSD=\sqrt{\frac{1}{N-2}\sum_{i=1}^{N-2}\left(RR_{i+1}-RR_{i}\right)},$$
where *N* is the number of detected R-peaks and $RR_{i}$ denotes the *i*th RR interval.

It is expected that the R-peak detection in noisy ECG signals will not work optimally. In these signals, false positives and false negatively detected peaks are expected. As a result, the tachogram will contain abnormal oscillations and the RMSSD from a clean ECG signal is expected to be lower compared to the one of a noisy recording. Therefore, a threshold is defined for a tolerable RMSSD value as $\mu_{RMSSD}+2*\sigma_{RMSSD}$. In this work, $\mu_{RMSSD}$ and $\sigma_{RMSSD}$ are chosen as 27 and 12, respectively. These values were selected based on [41]. If the absolute value of the difference between $RR_{i+1}$ and $RR_{i}$ is higher than this threshold, the *i*th RR interval is considered an abnormal one. Then, the number of abnormal RR intervals are used to calculate $SQI_{hrv}$ as:(2)$$SQI_{hrv}=1-\frac{total\;abnormal\;RR}{N-2},$$
which will be close to 0 for signals with very poor quality and close to 1 otherwise.

#### 3.2.4. Sqi Based on Ecg Morphology ($Sqi_{Qrs}$)

ECG morphology is particularly important in applications that not only focus on HR and HRV analysis but also the detection of cardiac events. Therefore, an SQI should reflect this by basing the output scores on the morphology of the ECG signals rather than on the HR. The $SQI_{avg}$, introduced in Section 3.2.1, also focuses on morphology and approaches with similar intent have been proposed in the literature, e.g., adaptive template matching [42]. However, both methods rely on calculating an average QRS complex template that can then be compared to each QRS complex. The calculation of the average template is particularly dependent on the R-peak detector performance and on the window size used. To reduce this dependency, we propose an alternative method that does not rely on an average template. Instead, the R-peaks of the ECG signals are used to extract the single QRS complexes. This is done using the *ecg_segment* method of the Neurokit Toolbox [35]. Afterwards, the QRS complexes in the current window are compared using the Pearson correlation coefficient. This comparison is performed by using consecutive QRS complexes. Therefore, each QRS complex will be compared with the following one. In particular, let $QRS_{i}$ represent the $i^{th}$ QRS complex in the current window. The correlation is estimated as:(3)$$\rho_{i}=\frac{\overset{L}{\underset{t=1}{\sum }}\left(QRS_{i}\left(t\right)\right)\left(QRS_{i+1}\left(t\right)\right)}{{\left(\sum_{t=1}^{L}{\left(QRS_{i}\left(t\right)\right)}^{2}\sum_{t=1}^{L}{\left(QRS_{i+1}\left(t\right)\right)}^{2}\right)}^{1/2}},$$
with *L* being the duration of the QRS complex. *N* is the number of R-peaks detected and, in each window, (N−1) comparisons will be performed between consecutive QRS complexes. The use of consecutive QRS complexes for the comparison is the main difference compared to approaches relying on an average template. The average correlation coefficient will be used as SQI for each window:(4)$$SQI_{QRS}=\frac{\sum_{i=1}^{N-1}\rho_{i}}{N-1}.$$

### 3.3. Sqi Comparison

Each SQI explained in Section 3.2 generates a different output. For this reason, we harmonize these outputs with the labels given in the QDB dataset. In Table 4, the results are mapped to the QDB. Since the output of the $SQI_{fuzzy}$ was classified into three options (quality levels), as was the QDB, the alignment was straightforward. For the remaining three SQIs, thresholds were empirically determined to map the numeric results to the annotation of the QDB. We based this decision on a previous work [36], where segments of ECG with quality >80 are considered high-quality level signals.

After harmonizing the labels, the performance of the SQIs was assessed in terms of the accuracy to classify the ECG segments in the categories shown in Table 4. The SQI with the highest accuracy in the QDB dataset was chosen as the best-performing one and it was applied to the signals from the clinical study.

### 3.4. Analysis of Quality Changes

The SQI selected in the QDB as the best-performing one was used to analyze the quality of the ECG collected from the cardiac patients recorded with the wearable health patch. The selected SQI’s performance was validated on ECG signals from healthy adults, whereas our clinical dataset contains data from cardiac patients. Therefore, we can expect differences in the performance of this SQI. We aimed to compensate for the difference in population with the preprocessing step to reject irregular QRS complexes.

The evaluation of the quality of the signals was based on different comparisons. In particular, the quality during full days, and daytime and night, was analyzed separately. Considering that the patients wore the patch for 5 days starting around 9:00–11:00 and that the time of removal was not exactly the same as the time of application, partial days are present in the data at the start and at the end. Figure 6 shows an example of a timeline and the separations in full day, daytime, and night. The following criteria were used to define the period of time to which the recording belonged:Daytime and night: the ECG data was split into daytime (assumed to be from 7:00 to 23:00) and night (assumed to be from 23:00 to 7:00). Boxplots were used to analyze the distribution of the quality of all the windows in the daytime and night moments.Full days: the average quality and the standard deviation of all the windows in 24 h were computed. The 24 h were calculated as the hours between 00:00 and 23:59.

A Wilcoxon–Mann–Whitney statistical test was used to gain insight into the statistical differences between the quality during daytime and night, and between the quality of the first day compared to the following days.

## 4. Results

The results obtained in this study are presented in this section.

### 4.1. Sqi Comparison

The confusion matrices of the SQIs compared using the QDB are shown in Figure 7a–d for $SQI_{avg}$, $SQI_{fuzzy}$, $SQI_{hrv}$, and $SQI_{QRS}$, respectively. The overall accuracies for $SQI_{avg}$ and $SQI_{fuzzy}$ were 37.70% and 55.04%, respectively. Both resulted in higher accuracy for the classification of “barely acceptable” compared to $SQI_{hrv}$ and $SQI_{QRS}$. The worst performing class for $SQI_{fuzzy}$ was “unacceptable”, with 0.14%. $SQI_{hrv}$ yielded an accuracy of 40.32%, with “unacceptable” outperforming the other classes. Finally, $SQI_{QRS}$ is the best-performing SQI with an overall accuracy of 76.69%. $SQI_{QRS}$ was, therefore, used for the analysis of the clinical dataset.

Given that $SQI_{avg}$ was the worst-performing method, we have included its results in Appendix A as reference for future work using this SQI in signals from subjects with CCS.

### 4.2. Analysis of Quality Changes

Changes in ECG signal quality were analyzed daily, distinguishing daytime and night. Figure 8 shows the distribution of the quality per minute during the days and nights as well as whether there is a statistical difference between day–night pairs per patient based on the Wilcoxon–Mann–Whitney test. The results showed that, of the day–night pairs among all the patients, only three of them were not significantly different: pairs Day 4–Night 4 and Day 5–Night 5 for patient 1 and pair Day 4–Night 4 for patient 4. In the remaining pairs, the null hypothesis was rejected with at least *p*-values ≤ 0.001. Furthermore, as visible in Figure 8, patients 4 and 5 presented the best quality in the dataset. Patient 3 had poor quality throughout most days with a median of four out of five days below 0.5. According to the *p*-value, the symbols described in Table 5 were used to denote the significant differences in the figures.

Changes in the quality of data by activities are identifiable even visually. Figure 9 shows ECG segments recorded by the Vital Signs patch in different settings. It can be seen that, when the quality is lower than 0.8 (Figure 9d–f), it is difficult to identify the complexes but when the quality is higher than 0.8 (Figure 9a–c) the complexes can be visualized.

Figure 10 shows the average quality and the standard deviation of the quality over full days for all the patients. Some patients, e.g., patient 3, seem to have a reduction of average quality during the days. To assess if the changes in quality are statistically significant the Wilcoxon–Mann–Whitney test was used.

In Table 6, the results of this statistical test between the average quality of the first day (used as baseline) and the average quality of the following days considering all the patients are presented. The null hypothesis is not rejected for all the days indicating that the difference in average quality during the days is not significant.

## 5. Discussion

Four SQIs have been compared in this work using the QDB dataset since quality annotations were available. $SQI_{QRS}$ was the best performing SQI, with not only the highest overall accuracy but also a more balanced performance in all the classes compared to the other three SQIs. It should be considered that the performance of $SQI_{hrv}$ could be improved with an adaptation of the number of standard deviations. However, considering our targeted subject group, cardiac patients, an SQI based on HRV may not be suitable due to the abnormal beats, given the fact that indexes derived from HRV are significantly affected by irregular heartbeats [43], which commonly occur in heart diseased populations [44]. Furthermore, HRV is known to change based on many variables, such as ageing, gender, or medication intake [45,46]. $SQI_{fuzzy}$ resulted in being too optimistic in the quality scores, misclassifying almost all the segments with unacceptable quality. $SQI_{avg}$, on the other hand, was the lowest performing SQI. Approaches based on an average template are implicitly assuming the overall quality of the segment is acceptable. However, depending on the used window size this may not be the case, for example, due to the presence of motion artifacts.

Considering the dataset recorded with the cardiac patients, it is worth mentioning that, as already mentioned in Section 2, data loss was present. There could be a link between the data loss and the activities performed by the patients. For example, in the case of patient 6, the data was not recorded after the third day, this could be due to poor electrode contact and motion artifacts since the patient performed three long cycling sessions for 174, 140, and 240 min on days 3, 4, and 6, respectively, as visible in Appendix A. However, further analysis and evaluations are required to reach a conclusion on this point.

The best performing SQI, $SQI_{QRS}$, was applied in the cardiac dataset. The SQIs may have different performances with cardiac patients since their ECG signals could include for example arrhythmia, premature ventricular contractions, or premature atrial contractions. A preprocessing step to discard irregular QRS complexes was indeed included to mitigate the effect that possible differences in ECG morphology were expected in this population. To further assess if the performance of the SQI is comparable between the two populations, quality annotations of the cardiac dataset are required. This should be considered in future works.

By comparing the distributions of the quality assessed per minute between daytime and night, differences can be observed. In Figure 8, the median of the quality during the night is consistently higher compared to the daytime. This was already expected as there were fewer motion artifacts and resting HR led to a better morphology [47]. Patient 3 is the only one with a median quality below 0.5 for most days (4 out of 5), along with the lowest quality during the nights. Considering the quality was already poor from day one, it is possible that the patch was not properly applied. Overall, the median quality for patient 6 is particularly low during the daytime compared to all the other patients but not during the night. As mentioned above, this could be associated with the amount of physical activity performed since, according to the diary visible in Appendix A, patient 6 had a total of 650 min of physical activity, being the second most active person after patient 3 (1070 min). In the statistical test performed to compare daytime and night, all patients had all or a majority of *p*-values ≤0.05. Therefore, the quality during the daytime is not directly associated with the quality at night and the difference between the quality in the two cases is statistically significant.

Figure 10 presents the average and the standard deviation of the quality during full days. Patient 3, as already mentioned, has the worst quality with the average quality in all the days being below 0.6. The average quality of both patients 2 and 3 decreases after the third day. The Wilcoxon–Mann–Whitney test was performed to assess whether there is a significant difference between the average qualities of the baseline (day 1) and the average qualities of days 2–6 for all the patients. The results suggest there is no statistically significant change in quality over time. Therefore, even if there are changes in quality during the days, these changes are not statistically significant in all patients.

This study can be considered a first pilot for the use of the *Vital Signs patch* in cardiac patients. The use of wearable technologies in this population could lead to more accurate CVD screening and diagnosis. However, differences in the ECG morphology in this population compared to the healthy subjects group need to be expected. By analyzing patients, we can find the activities that degrade ECG signal quality (such as biking) and use this to find future solutions from experimental setup to data processing. Therefore, SQIs, which represent an important initial step for ECG signals evaluation before further processing (e.g., automated cardiac events detection), should be further developed taking into account the targeted population. Quality annotations on ECG collected on cardiac patients are required to further develop in this direction.

## 6. Conclusions

This study analyzed the changes in ECG signal quality of a (*Vital Signs patch* platform featuring) self-adhesive dry electrodes technology for long-term ECG monitoring, looking for significant quality differences over days–nights and over time. No significant difference was found in the quality changes between the days. Therefore, the self-adhesive dry electrodes could be a promising technology for long-term ECG monitoring. Future studies shall assess the benefits of this technology in more detail. Particular attention shall be paid to motion artifacts and physical activities as they seem to be linked to quality changes as shown by the reduced quality during daytime compared to night. Future developments of SQIs for ECG should address the limitations of this pilot study including small panel size. Likewise, our future work consists of analyzing the quality of signals in segments where CVD were identified.

## Figures and Tables

**Figure 1 sensors-23-02130-f001:**
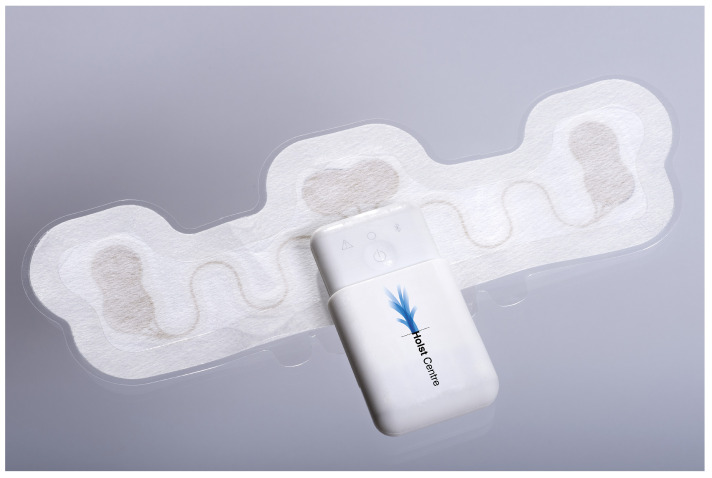
Single-lead *Vital Signs patch* exploded view.

**Figure 2 sensors-23-02130-f002:**
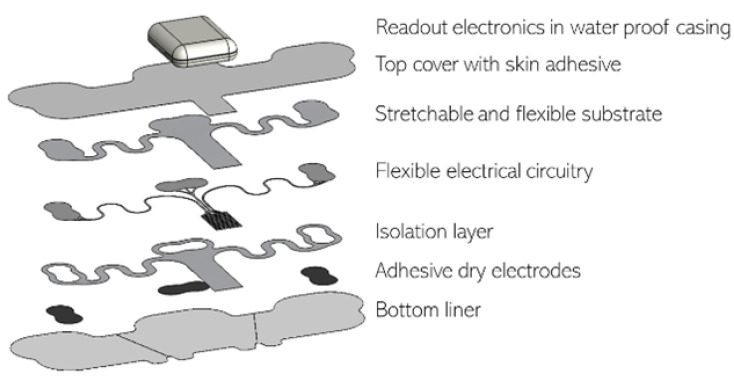
Single-lead *Vital Signs patch* based on self-adhesive dry electrodes.

**Figure 3 sensors-23-02130-f003:**
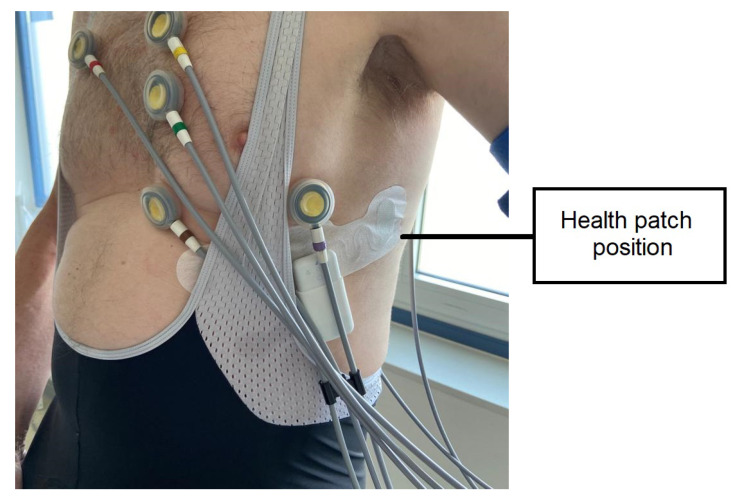
Cardiac patient during the EST wearing standard 12-lead ECG electrodes and the *Vital Signs patch* in the 5th intercostal space.

**Figure 4 sensors-23-02130-f004:**
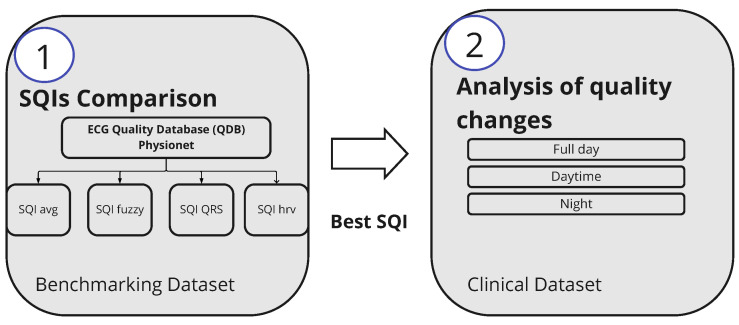
Summary of the processing steps.

**Figure 5 sensors-23-02130-f005:**
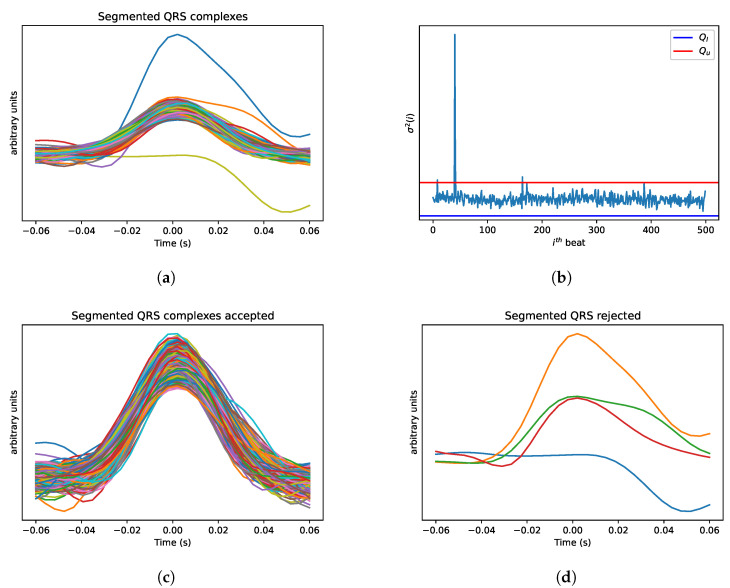
Example results of preprocessing method to reject irregular QRS complexes. (**a**) Segmented QRS complexes. (**b**) Segmented QRS complexes variance distribution and confidence intervals. (**c**) Accepted segmented QRS complexes. (**d**) Rejected segmented QRS complexes. Signal colours are arbitrary to keep the segmented QRS complexes identifiable.

**Figure 6 sensors-23-02130-f006:**
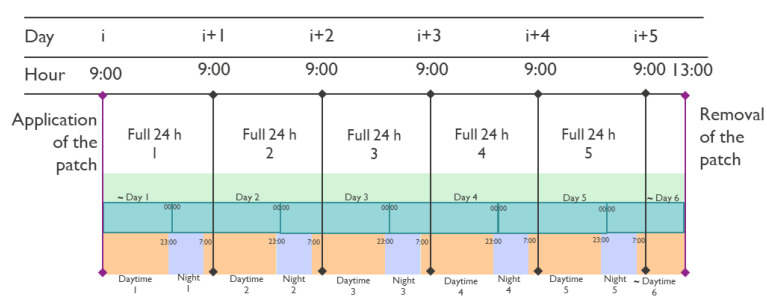
Example protocol timeline and separations used in the analysis. The ∼ symbol indicates the partial full days and daytime.

**Figure 7 sensors-23-02130-f007:**
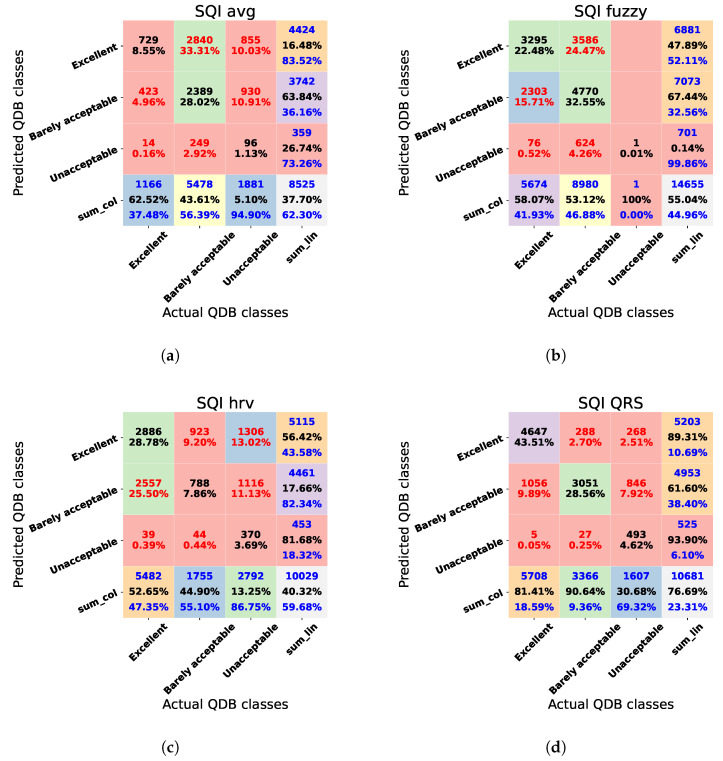
Confusion matrices for the SQIs compared using the QDB dataset. (**a**) Confusion matrix for $SQI_{avg}$. (**b**) Confusion matrix for $SQI_{fuzzy}$. (**c**) Confusion matrix for $SQI_{hrv}$. (**d**) Confusion matrix for $SQI_{QRS}$.

**Figure 8 sensors-23-02130-f008:**
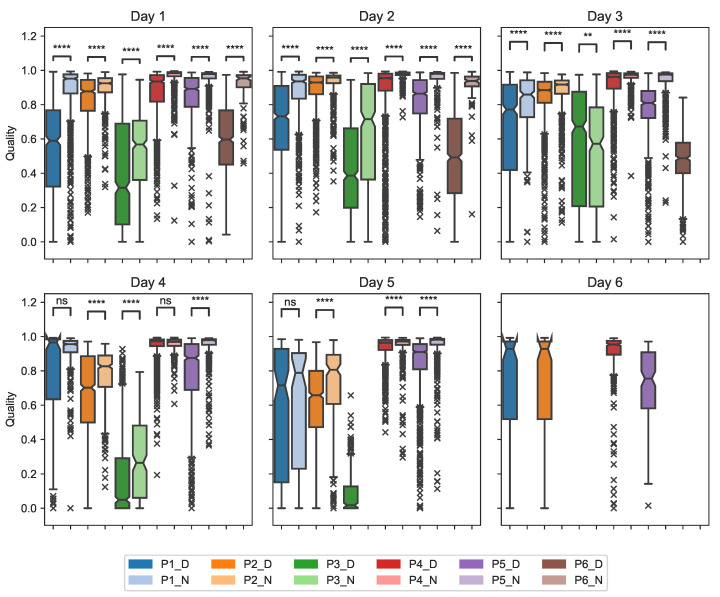
Distribution and statistical analysis of ECG quality during daytime and night for all patients.. The explanation of *, **, ***, **** and ns correspond to the definitions of Table 5.

**Figure 9 sensors-23-02130-f009:**
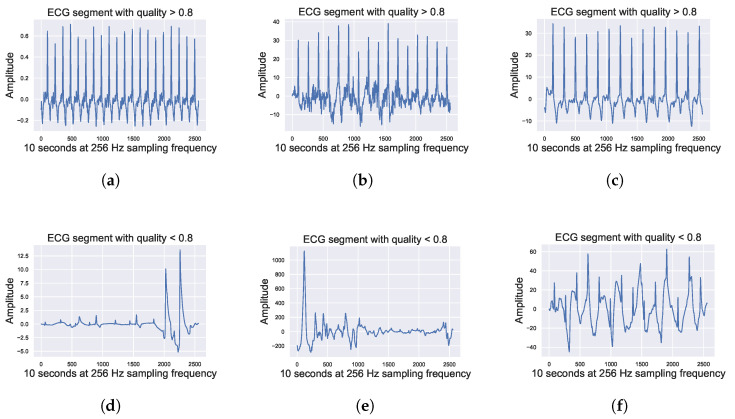
ECG segments of recording segments by Vital Signs patch in different settings (during the day, night, and exercise) with quality higher and lower than 0.8. (**a**) ECG segment during the stress test, quality signal >0.8. (**b**) ECG segment during the day, quality signal >0.8. (**c**) ECG segment during the night, quality signal >0.8. (**d**) ECG segment during the stress test, quality signal <0.8. (**e**) ECG segment during the day, quality signal <0.8. (**f**) ECG segment during the night, quality signal <0.8.

**Figure 10 sensors-23-02130-f010:**
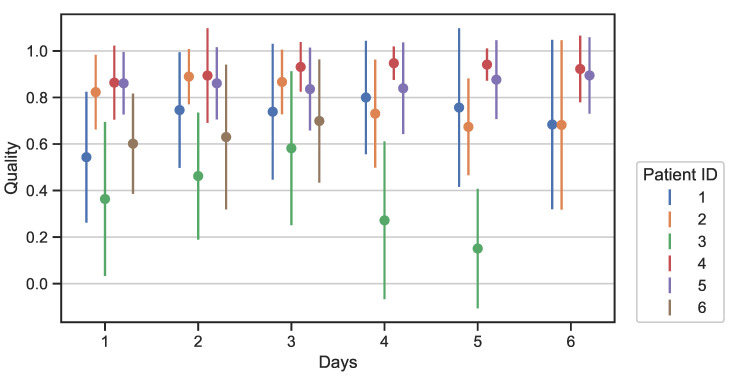
Data quality over time for all patients. Some patients have fewer days due to data loss.

**Table 1 sensors-23-02130-t001:** Demographics of the 15 subjects in the QDB.

Characteristic	Values (n = 15)
Age (years)	40.6 ± 19.64
Male (n,%)	6 (40)
BMI (kg/m^2^)	22.82 ± 3.98

**Table 2 sensors-23-02130-t002:** Cardiac patient characteristics.

Characteristic	Values (n = 6)
Age (years)	69.8 ± 6.2
Male (n, %)	6 (100)
BMI (kg/mm^2^)	25.3 ± 1.8

**Table 3 sensors-23-02130-t003:** Available data in hours, split also into daytime (assumed from 7 a.m. to 11 p.m.) and night (assumed from 11 p.m. to 7 a.m.). The last column contains the HR max reached during the stress test by each patient. NA stands for Not Applicable.

Patient ID	Usable Data (Hours)			HR Max
2–4	*Total*	*Daytime*	*Night*	
*1*	118.8	78.9	39.9	151
*2*	117.03	78.03	39.01	96
*3*	99.0	67.2	31.8	137
*4*	119.97	80.08	39.9	127
*5*	119.6	79.7	39.9	166
*6*	52.8	36.8	16.0	139
**Cumulative**	627.19	420.71	206.51	*NA*

**Table 4 sensors-23-02130-t004:** Aligning pipeline outputs with QDB labels for evaluation.

QDB	${\mathrm{SQI}}_{\mathrm{fuzzy}}$	${\mathrm{SQI}}_{\mathrm{avg}},\;{\mathrm{SQI}}_{\mathrm{hrv}},\;{\mathrm{SQI}}_{\mathrm{QRS}}$
Class 0	Excellent	SQI ≥0.8
Class 1	Barely Acceptable	0.5≤ SQI <0.8
Class 2	Unacceptable	SQI <0.5

**Table 5 sensors-23-02130-t005:** Symbols used to report statistical significance.

Symbol	Meaning
ns	*p*-value > 0.05
*	*p*-value ≤ 0.05
*	*p*-value ≤ 0.01
**	*p*-value ≤ 0.001
***	*p*-value ≤ 0.0001

**Table 6 sensors-23-02130-t006:** Wilcoxon–Mann–Whitney test between baseline (first day) and the following days, among all the patients.

*Day*	*2*	3	4	5	6
*p*-value	0.68571	0.4857	1.0	0.48571	0.88571

## Data Availability

Data cannot be shared due to a confidentiality agreement in the project.

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
