# Peer review of "Signal Quality Analysis for Long-Term ECG Monitoring Using a Health Patch in Cardiac Patients"

_sensors, 2023, doi:10.3390/s23042130_

Round 1

Reviewer 1 Report

The authors propose to estimate the long-term quality of the ECG obtained with a patch. The analysis focuses on subjects with chronic coronary syndrome (CCS) using signal quality indices.

The work is well presented, and the methodology is well described, however, the results are presented in a confusing way.

Here are my comments:

1. In the SQI(QRS) calculation, the authors first estimate the Pearson correlation coefficient. They comment that the QRS complexes of the current window are compared. My question is what do you compare it to? With the original signal? On the other hand, does the correlation index depend on the size of the window? I consider it important to clarify this point, and I think it would be useful to present a table with the results of the correlation coefficients (from equation (3)) for each one of the subjects since it is obvious that the correlation coefficient defines SQI(QRS ).

2. The authors used a dataset for benchmarking, where ECG signals were acquired with a sampling frequency of 1 kHz. In the case of the patch, a sample rate of 256 Hz was used. I wonder if the difference in signal sampling (which defines the number of samples in the windows) doesn't affect the estimate of SQIs that are based on the morphology of the signal, i.e.: SQI(AVG) and SQI(QRS). I think it is important to clarify this point.

3. From the confusion matrices, the authors conclude that SQI(QRS) is the indicator they will use to assess ECG quality in subjects with CCS. I understand that it is the best case. However, why didn't they use the worst-case SQI(AVG) to see the results it returns? From my point of view, this would be helpful for future works where this type of indicator is used to assess the ECG of subjects with CCS.

4. In section 2.2.2, the authors comment that the EST lasted until the maximum heart rate was reached. What is the maximum heart rate? Is it the same for all subjects? This point is confusing, so it should be better described.

5. Did the test subjects sign any informed consent? I think it is important to describe the ethical statements in a section at the end of the manuscript.

 Minor comments:

1. In table 1, n = 18. If the male subjects were 8, I think the percentage value is 44.44%

2. In Table 1, n = 18, but in section 2.1, the authors comment that there were 6 men and 9 women (n=15).

3. In figure 5, use another symbol other than "*", since it could be confused with the symbols defined in table 5.

4. In Figure 7, day 6, the upper marks (ns, ***) do not appear. Any reason?

5. In Table 7, there is a typo in the first row, patient 1: stess instead of stress.

Reviewer 2 Report

In this pilot study, the authors analyzed the usability of a Vital Signs patch platform featuring self-adhesive dry electrodes technology for long-term ECG monitoring of cardiac patients. The authors found no significant difference in the quality changes between the days, however, a higher quality during the nights could be observed, confirming the link with motion artifacts.

This study addresses a current topic regarding ambulatory patient monitoring over several days, especially of patients with cardiovascular disease.

1.  To facilitate understanding, the authors should include and compare a few examples of ECG recording segments by Vital Signs patch in different settings (e.g. during day, night, exercise) in their manuscript.

2. Can the authors provide any information about patient`s impression concerning the worn Vital Signs patch?

3. Signal quality plays a crucial role in patient monitoring, especially in ambulatory setting. Beside motion artifacts, a loss of signal quality can often be observed in case of cardiac arrhythmias, hampering detection and diagnosis of the latter. In the Discussion, the authors state that “This study can be considered a first pilot for the use of the Vital Signs patch in cardiac patients. The use of wearable technologies in this population could lead to more accurate CVD screening and diagnosis”.

 - Are any information available regarding the occurrence and the type of      cardiac arrhythmias in this patient population? If cardiac arrhythmias have been detected, can the authors draw any conclusions about the reliability of Vital Signs patch in the single cases? This important aspect should be included in the manuscript. Besides, the authors should elucidate which findings in their study support the usability of a Vital Signs patch for long-term ECG monitoring especially of cardiac patients. 

Reviewer 3 Report

The article presents the algorithms that allow to analyze the ECG signal with the long-term monitoring using a health patch. The subject matter presented in the article is current and should be developed with the increasing number of cardiological diseases. The issues presented in the article make it possible to understand the problem, but sometimes they should be detailed. The research methodology described in the work contains the required information, as well as the research area. The work is well organized, with an introduction, the theoretical part, the application part and the conclusions. However, there is a noticeable random arrangement of the figures and tables, which should be corrected.

Recommendations for improving the manuscript:

1. Is it worth writing the summary conclusions (last 3 sentences) in the summary?

2. It is a pity that the analysis was conducted for such a small number of patients (6 people).

3. The arrangement of the figures and the tables is accidental - e.g. Fig.1 is referred to in section 2.2.1, and it is presented in section 3.1. This arrangement makes it difficult to read the text.

4. The font in figures 5 and 6 is hard to read.

5. Figure 4.a lacks a legend explaining which segmented QRS complexes are normal and which are irregular.

6. What is the reason for placing the table 7 at the end of the text, and not in the text like other tables?
